# Custom-Design of FDR Encodings: The Case of Red-Black Planning

**Daniel Fišer**[1,2]**, Daniel Gnad**[1]**, Michael Katz**[3]**, Jörg Hoffmann**[1]

[1] Saarland University, Saarland Informatics Campus, Saarbrücken, Germany
[2] Czech Technical University in Prague, Faculty of Electrical Engineering, Czech Republic
[3] IBM Research, Yorktown Heights, NY, USA
danfis@danfis.cz, {gnad,hoffmann}@cs.uni-saarland.de, michael.katz1@ibm.com

## Abstract

Classical planning tasks are commonly described in PDDL, while most planning systems operate on a grounded finite-domain representation (FDR). The translation of PDDL into FDR is complex and has a lot of choice points—it involves identifying so called mutex groups—but most systems rely on the translator that comes with Fast Downward. Yet the translation choice points can strongly impact performance. Prior work has considered optimizing FDR encodings in terms of the number of variables produced. Here we go one step further by proposing to custom-design FDR encodings, optimizing the encoding to suit particular planning techniques. We develop such a custom design here for red-black planning, a partial delete relaxation technique. The FDR encoding affects the causal graph and the domain transition graph structures, which govern the tractable fragment of red-black planning and hence affects the respective heuristic function. We develop integer linear programming techniques optimizing the scope of that fragment in the resulting FDR encoding. We empirically show that the performance of red-black planning can be improved through such FDR custom design.

## Introduction

Classical planning tasks are usually defined in PDDL (McDermott 2000), which is a lifted representation based on first-order logic. Yet, most state-of-the-are planners, in particular the systems based on Fast Downward (FD) (Helmert 2006), use a *finite-domain representation (FDR)* encoding, where subsets of propositional facts are grouped into finite-domain state variables. This is possible only if reachable states make at most one of the grouped facts true, i. e., if they form a *mutex group*. FDR encodings are useful for a wide variety of purposes like abstraction based heuristics (Haslum et al. 2007; Katz and Domshlak 2010; Helmert et al. 2014; Seipp and Helmert 2018), or red-black planning heuristics (Domshlak, Hoffmann, and Katz 2015; Gnad and Hoffmann 2015; Katz 2019), where a subset of (red) variables are delete-relaxed while the rest (black) keep their true semantics.

The translation of PDDL into FDR is complex—deciding whether a set of facts forms a mutex group is as hard as planning itself. There are many choice points in identifying mutex groups and grouping facts into state variables. Systems based on Fast Downward almost universally rely on its translator (Helmert 2009) and do not question the choices made there.

Yet the translation choices can impact performance. In prior work (Dvořák, Toropila, and Barták 2013, 2015), a minimization of the number of FDR variables was proposed assuming fewer variables result in more information being aggregated into them. The results show that significantly reducing the number of variables can improve the performance of FDR-based techniques. The improvements are moderate though, and there is a lot of variance depending on domain and planning technique; a consistent picture does not emerge.

Here we show that this picture can change when going one step further in the target of FDR optimization: we propose to *custom-design FDR encodings, optimizing the encoding to suit particular planning techniques.*[1] We develop such a custom design for *red-black planning*, which distinguishes between relaxed (red) and non-relaxed (black) variables. We focus on a polynomial-time solvable fragment of red-black planning (Katz, Hoffmann, and Domshlak 2013b) which uses a heuristic function based on generating a red-black plan for a given state (Katz, Hoffmann, and Domshlak 2013a). The tractability of red-black planning is determined by the decision which variables are painted red or black. The state-of-the-art approach uses *painting strategies* (Domshlak, Hoffmann, and Katz 2015) assuming the FDR encoding is given and fixed, and decides only on variable colors. Here, we open up this design space and allow to custom-design the FDR variables and their painting at the same time.

The tractability of red-black planning is governed by the FDR encoding's *causal graph* (whose projection onto the black variables must be acyclic) and *domain transition graphs* (that must be invertible for the black variables). We develop techniques optimizing these explicit syntactic criteria, to be able to choose from different viable FDR encodings of tractable red-black planning tasks. The optimization itself is then formulated as an integer linear program, where the allocation of black (non-relaxed) variables is controlled

---

[1] Vallati et al. (2015; 2017) have explored domain model optimization tailored to specific planners before, but for PDDL not for FDR variable design, and changing only the ordering of artefacts in the input model (which had previously been observed to be relevant for the performance of some planners (Howe and Dahlman 2002)).

by the program's objective function.

## Background

A **STRIPS planning task** $\Pi$ is specified by a tuple $\Pi = \langle \mathcal{F}, \mathcal{O}, s_I, s_G \rangle$, where $\mathcal{F} = \{f_1, \ldots, f_n\}$ is a set of facts, and $\mathcal{O} = \{o_1, \ldots, o_m\}$ is a set of grounded operators. A **state** $s \subseteq \mathcal{F}$ is a set of facts, $s_I \subseteq \mathcal{F}$ is an **initial state** and $s_G \subseteq \mathcal{F}$ is a **goal** specification. A state $s$ is a **goal state** iff $s_G \subseteq s$. An **operator** $o$ is a tuple $o = \langle \text{pre}(o), \text{add}(o), \text{del}(o) \rangle$, where $\text{pre}(o) \subseteq \mathcal{F}$ is the set of preconditions of $o$, and $\text{add}(o) \subseteq \mathcal{F}$ and $\text{del}(o) \subseteq \mathcal{F}$ are the sets of add and delete effects, respectively. All operators are well-formed, i.e., $\text{add}(o) \cap \text{del}(o) = \emptyset$ and $\text{pre}(o) \cap \text{add}(o) = \emptyset$. We assume unit costs. An operator $o$ is **applicable** in a state $s$ if $\text{pre}(o) \subseteq s$. The **resulting state** of applying an applicable operator $o$ in a state $s$ is the state $o[\![s]\!] = (s \setminus \text{del}(o)) \cup \text{add}(o)$.

A sequence of operators $\pi = \langle o_1, \ldots, o_n \rangle$ is applicable in a state $s_0$ if there are states $s_1, \ldots, s_n$ such that $o_i$ is applicable in $s_{i-1}$ and $s_i = o_i[\![s_{i-1}]\!]$ for $1 \leq i \leq n$. The resulting state of this application is $\pi[\![s_0]\!] = s_n$. A sequence of operators $\pi$ is called a **plan** iff $\pi$ is applicable in $s_I$ and $s_G \subseteq \pi[\![s]\!]$.

A set of facts $M \subseteq \mathcal{F}$ is called a **mutex** if $M \not\subseteq s$ for every reachable state $s$. A set of facts $M \subseteq \mathcal{F}$ is called a **mutex group** if $|M \cap s| \leq 1$ for every reachable state $s$, and $M$ is called a **fact-alternating mutex group** (fam-group) if $|M \cap s_I| \leq 1$ and $|M \cap \text{add}(o)| \leq |M \cap \text{pre}(o) \cap \text{del}(o)|$ for every operator $o \in \mathcal{O}$. Every fam-group is a mutex group. Every subset of a mutex group is a mutex group, but not every subset of a fam-group is a fam-group. Facts from every mutex group form pairwise mutexes (Fišer and Komenda 2018).

Given a set of mutex groups $\mathcal{G}$, $\mathcal{M}_\mathcal{G} \subseteq 2^\mathcal{F}$ denotes an upper set of mutexes induced by $\mathcal{G}$, i.e., for every mutex group $G \in \mathcal{G}$ and every $f, f' \in G$, $f \neq f'$, it holds that $\{f, f'\} \in \mathcal{M}_\mathcal{G}$, and for every $M \in \mathcal{M}_\mathcal{G}$ and every $f \in \mathcal{F}$ it holds that $M \cup \{f\} \in \mathcal{M}_\mathcal{G}$.

An **FDR planning task** $\mathcal{P}$ is specified by a tuple $\mathcal{P} = \langle \mathcal{V}, \mathcal{O}, \psi_I, \psi_G \rangle$. $\mathcal{V}$ is a finite set of **variables**, each variable $V \in \mathcal{V}$ has a finite **domain** $\text{dom}(V)$. A **fact** $\langle V, v \rangle$ is a pair of a variable $V \in \mathcal{V}$ and one of its values $v \in \text{dom}(V)$. A **partial state** $p$ is a variable assignment over some variables $\text{vars}(p) \subseteq \mathcal{V}$. We write $p[V]$ for the value assigned to the variable $V \in \text{vars}(p)$ in the partial state $p$. Given a set of variables $U \subseteq \mathcal{V}$, $p[U]$ denotes a partial state $p$ restricted to $U$. We also identify $p$ with the set of facts contained in $p$, i.e., $p = \{\langle V, p[V] \rangle \mid V \in \text{vars}(p)\}$. A partial state $s$ is a **state** if $\text{vars}(s) = \mathcal{V}$. $\psi_I$ is an **initial state**. $\psi_G$ is a partial state called **goal**, and a state $s$ is a **goal state** iff $\psi_G \subseteq s$. $\mathcal{O}$ is a finite set of **operators**, each operator $o \in \mathcal{O}$ has a precondition $\text{pre}(o)$ and effect $\text{eff}(o)$, which are partial states. An operator $o$ is **applicable** in a state $s$ iff $\text{pre}(o) \subseteq s$. The **resulting state** of applying an applicable operator $o$ in a state $s$ is the state $o[\![s]\!]$ where $o[\![s]\!][V] = \text{eff}(o)[V]$ for every $V \in \text{vars}(\text{eff}(o))$, and $o[\![s]\!][V] = s[V]$ for every $V \in \mathcal{V} \setminus \text{vars}(\text{eff}(o))$. Operator sequences and plans are defined analogously to STRIPS.

The **domain transition graph** (DTG) of a variable $V$, denoted by $\mathcal{D}_V$, is an edge-labeled multi-digraph with vertices $\text{dom}(V)$ and with an edge from $d$ to $d'$ induced by an operator $q \in \mathcal{O}$ and denoted by $(d, q, d')$ iff $d \neq d'$, $\text{eff}(q)[V] = d'$, and either $\text{pre}(q)[V] = d$ or $V \notin \text{vars}(\text{pre}(q))$.

Let $\mathcal{U} \subseteq \mathcal{V}$ denote a subset of variables. The **causal graph** $\text{CG}_\mathcal{U}$ of $\mathcal{U}$ is a digraph with vertices $\mathcal{U}$. An edge $(V, V')$ is in $\text{CG}_\mathcal{U}$ iff $V \neq V'$ and there exists an operator $o \in \mathcal{O}$ such that $(V, V') \in (\text{vars}(\text{pre}(o)) \cup \text{vars}(\text{eff}(o))) \times \text{vars}(\text{eff}(o))$.

A **red-black planning task** $\Pi$ is specified by a tuple $\Pi = \langle \mathcal{V}^R, \mathcal{V}^B, \mathcal{O}, \psi_I, \psi_G \rangle$, where $\mathcal{V}^R$ and $\mathcal{V}^B$ are state variables, called **red variables** and **black variables**, respectively. Black variables have the value-switching semantics as in FDR. Red variables have the value-accumulating semantics, i.e., operators "extend" the value of $V \in \mathcal{V}^R$ from $\{x\}$ to $\{x, y\}$. Applicability of operators, sequences of operators, and red-black plans are defined accordingly [see Domshlak, Hoffmann, and Katz, 2015].

Given a variable $V \in \mathcal{V}$, an edge $(d, q, d')$ from $\mathcal{D}_V$ is **relaxed side effects invertible** (RSE-invertible) if there exists an edge $(d', q', d)$ such that $\text{pre}(q')[\mathcal{V} \setminus V] \subseteq \text{pre}(q)[\mathcal{V} \setminus V] \cup \text{eff}(q)[\mathcal{V} \setminus V]$. $V$ is RSE-invertible if every edge in $\mathcal{D}_V$ is RSE-invertible.

RSE-invertibility is a sufficient criterion for tractability in the case when the subgraph of the causal graph restricted to the black variables is acyclic (Katz, Hoffmann, and Domshlak 2013a). The actual algorithm presented in that work handles only the arc-less black causal graph case. Later, an algorithm for devising red-black plans for the acyclic causal graph case was presented (Katz and Hoffmann 2014; Domshlak, Hoffmann, and Katz 2015). In what follows, we focus on the **red-black heuristic** $h^{\text{RB}}$ for a state $s$, that returns the length of a red-black plan for $s$, computed for that tractable fragment.

## Translation from STRIPS to FDR

A concise translation from STRIPS to FDR requires mutex groups that are used for creating FDR variables from STRIPS facts. Before we formalize the translation process, we formulate an intermediate STRIPS representation where we assume the mutex groups are already inferred and the STRIPS planning task is pruned with the mutex information contained in them. Moreover, we need to preserve the DTG structure of mutex groups from which black variables are created to preserve their RSE-invertibility. For this purpose, we introduce a mapping from mutex groups to fam-groups containing these mutex groups, called *fam-group map*. We choose specifically fam-groups because of their simpler structure (Fišer and Komenda 2018). It is not a limitation, since fam-groups are the most commonly used type of mutex groups anyway (Fišer 2020).

**Definition 1.** Given a set of mutex groups $\mathcal{G}$, $\gamma : \mathcal{G} \mapsto 2^\mathcal{F}$ is a **fam-group map** if for every $G \in \mathcal{G}$ it holds that either $\gamma(G) = \emptyset$ or $\gamma(G)$ is a fam-group such that $G \subseteq \gamma(G)$.

**Definition 2.** Given a STRIPS planning task $\Pi = \langle \mathcal{F}, \mathcal{O}, s_I, s_G \rangle$, and a set of non-empty mutex groups $\mathcal{G}$, and a fam-group map $\gamma$ over $\mathcal{G}$, $\text{E}_\Pi = \langle \Pi, \mathcal{G}, \gamma \rangle$ denotes

a **STRIPS planning task extended with mutex groups** (MGE-STRIPS) if all of the following hold:

(S1) $\mathcal{F} = \bigcup_{o \in \mathcal{O}} \mathrm{add}(o) \cup \mathrm{del}(o)$,

(S2) for every operator $o \in \mathcal{O}$ it holds that $\mathrm{pre}(o) \notin \mathcal{M}_\mathcal{G}$ and $o[\![\mathrm{pre}(o)]\!] \notin \mathcal{M}_\mathcal{G}$ and for every $f \in \mathrm{del}(o)$ it holds that $\{f\} \cup \mathrm{pre}(o) \notin \mathcal{M}_\mathcal{G}$,

(S3) $s_I \notin \mathcal{M}_\mathcal{G}$ and $s_G \notin \mathcal{M}_\mathcal{G}$,

(S4) $\bigcup_{G \in \mathcal{G}} G = \mathcal{F}$,

(S5) for every $G, G' \in \mathcal{G}$, $G \neq G'$ it holds that $G \cap G' = \emptyset$,

(S6) $\{f\} \in \mathcal{G}$ for every $f \in \bigcup_{o \in \mathcal{O}} (\mathrm{del}(o) \setminus \mathrm{pre}(o))$,

(S7) $\gamma(G) \neq \emptyset$ for every $G \in \mathcal{G}$ such that $|G| \geq 2$.

Condition (S1) makes sure that there are no "static" facts, i.e., facts that either do not appear in any reachable state, or that are set in the initial state and remain set in all reachable states. (S2) corresponds to pruning of unreachable operators and delete effects that are not applicable. (S3) can be false only for unsolvable tasks. (S4) ensures that the set of mutex groups cover all facts. Pairwise disjoint mutex groups (S5) ensure that each STRIPS fact is encoded only once. (S6) ensures a polynomial translation without conditional effects (for a detailed explanation see (Helmert 2009, Section 7.3)). The fam-group $\gamma$ is used for encoding preconditions of operators in the translation to FDR, and, as we described before, we assume all mutex groups $\mathcal{G}$ to be based on fam-groups, i.e., each mutex group from $\mathcal{G}$ is either a singleton or it is mapped to some fam-group (S7).

Our formulation of the translation from MGE-STRIPS to an FDR planning task differs from the translation described by Helmert (2009) in the way preconditions of operators are encoded. We use the value $\perp_G$ to express that none of the facts from the mutex group $G$ is set. Whenever the fam-group map $\gamma$ maps $G$ to a non-empty fam-group and $\mathrm{pre}(o) \cap (\gamma(G) \setminus G) \neq \emptyset$, we set the variable $V_G$ to $\perp_G$ in a precondition of operator $q_o$, i.e., $V_G$ is set to $\perp_G$ only if $\mathrm{pre}(o)$ is mutex with $G$ and we can infer this fact from the given fam-group map. Note also that if $\gamma(G) = \emptyset$ for every $G \in \mathcal{G}$, then we get the same FDR encoding as Helmert (2009).

**Definition 3.** Given an MGE-STRIPS planning task $\mathrm{E}_\Pi = \langle \Pi = \langle \mathcal{F}, \mathcal{O}, s_I, s_G \rangle, \mathcal{G}, \gamma \rangle$, $\mathcal{P}(\mathrm{E}_\Pi) = \langle \mathcal{V}, \mathcal{O}, \psi_I, \psi_G \rangle$ is an FDR planning task such that:

(T1) $\mathcal{V} = \{V_G \mid G \in \mathcal{G}\}$ where $\mathrm{dom}(V_G) = G \cup \{\perp_G\}$ for every $V_G \in \mathcal{V}$;

(T2) $\mathcal{O} = \{q_o \mid o \in \mathcal{O}\}$ where

  (T2a) $\mathrm{pre}(q_o) = \{\langle V_G, f \rangle \mid G \in \mathcal{G}, G \cap \mathrm{pre}(o) \neq \emptyset, f \in G \cap \mathrm{pre}(o)\} \cup \{\langle V_G, \perp_G \rangle \mid G \in \mathcal{G}, G \cap \mathrm{pre}(o) = \emptyset, \gamma(G) \cap \mathrm{pre}(o) \neq \emptyset\}$,

  (T2b) $\mathrm{eff}(q_o) = \{\langle V_G, f \rangle \mid G \in \mathcal{G}, G \cap \mathrm{add}(o) \neq \emptyset, f \in G \cap \mathrm{add}(o)\} \cup \{\langle V_G, \perp_G \rangle \mid G \in \mathcal{G}, G \cap \mathrm{del}(o) \neq \emptyset, G \cap \mathrm{add}(o) = \emptyset\}$,

(T3) $\psi_I = \{\langle V_G, f \rangle \mid G \in \mathcal{G}, G \cap s_I \neq \emptyset, f \in G \cap s_I\} \cup \{\langle V_G, \perp_G \rangle \mid G \in \mathcal{G}, G \cap s_I = \emptyset\}$

(T4) $\psi_G = \{\langle V_G, f \rangle \mid G \in \mathcal{G}, G \cap s_G \neq \emptyset, f \in G \cap s_G\}$

Although encoding $\perp_G$ in preconditions of operators is not necessary, because this fact is true implicitly, we use

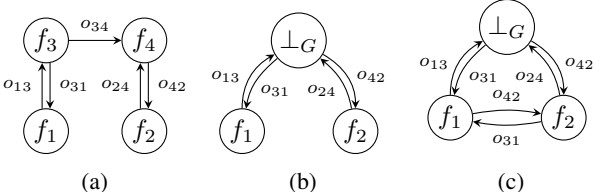

(a)  (b)  (c)

Figure 1: $F = \{f_1, f_2, f_3, f_4\}$ is a fam-group and $G = \{f_1, f_2\}$ a mutex group. (a) is a projection to $F$; (b) a DTG of $V_G$ if $\perp_G$ is encoded in operator preconditions, i.e., if $\gamma(G) = F$; (c) a DTG of $V_G$ if $\perp_G$ is *not* encoded in operator preconditions, i.e., if $\gamma(G) = \emptyset$.

it to ensure that the structure of a projection to the mutex group in STRIPS is preserved in the corresponding DTG in FDR. Why we need this will become clear in the next section where we deal with RSE-invertibility. For now, see the example depicted in Figure 1. The projection to the fam-group $F = \{f_1, \ldots, f_4\}$ is shown in Figure 1a. Now suppose we decide to encode only a subset of $F$ as an FDR variable, namely $G = \{f_1, f_2\}$. If we set $\gamma(G) = \emptyset$, then the resulting DTG of $V_G$ will be as depicted in Figure 1c, i.e., it will have additional edges between $f_1$ and $f_2$ not present in the original projection, because the operators $o_{31}$ and $o_{42}$ have $f_3$ and $f_4$, respectively, in their preconditions. Therefore $V_G$ is not set in $\mathrm{pre}(q_{o_{31}})$ and $\mathrm{pre}(q_{o_{42}})$. However, if we set $\gamma(G) = F$, then we preserve the structure, as depicted in Figure 1b, because $V_G$ will be set to $\perp_G$ in the preconditions of $q_{o_{31}}$ and $q_{o_{42}}$.

**Proposition 4.** *Let* $\mathrm{E}_\Pi = \langle \Pi = \langle \mathcal{F}, \mathcal{O}, s_I, s_G \rangle, \mathcal{G}, \gamma \rangle$ *denote an MGE-STRIPS planning task.* $\pi = \langle o_1, \ldots, o_n \rangle$ *is a plan for* $\mathrm{E}_\Pi$ *iff* $\pi' = \langle q_{o_1}, \ldots, q_{o_n} \rangle$ *is a plan for* $\mathcal{P}(\mathrm{E}_\Pi)$.

*Proof Sketch.* $\mathcal{P}(\mathrm{E}_\Pi)$ is well-defined, because (T3) assigns one value to every variable, i.e., $\psi_I$ is a state. Similarly, (T4) ensures that $\psi_G$ is a partial state, and (T2a-b) produce partial states, because only one value is assigned to each variable.

Note that $f \in s_I$ iff $\langle V_G, f \rangle \in \psi_I$ for every $G$ s.t. $G \cap s_I = \{f\}$, and $\langle V_G, \perp_G \rangle \in \psi_I$ only if $s_I \cap G = \emptyset$. Similarly, $f \in s_G$ iff $\langle V_G, f \rangle \in \psi_G$; and for every $o \in \mathcal{O}$ it holds that $f \in \mathrm{pre}(o)$ iff $\langle V_G, f \rangle \in \mathrm{pre}(q_o)$, and $\langle V_G, \perp_G \rangle \in \mathrm{pre}(q_o)$ only if $\mathrm{pre}(o) \cap G = \emptyset$ and $\mathrm{pre}(o) \cap \gamma(G) \neq \emptyset$. Hence $o$ is applicable in $s_I$ iff $q_o$ is applicable in $\psi_I$ for every $o \in \mathcal{O}$. Furthermore, from (T2b) it follows that $f \in o[\![s_I]\!]$ iff $\langle V_G, f \rangle \in q_o[\![\psi_I]\!]$ for every $G$ s.t. $G \cap o[\![s_I]\!] = \{f\}$, and $\langle V_G, \perp_G \rangle \in q_o[\![\psi_I]\!]$ only if $G$ is mutex with $o[\![s_I]\!]$. The rest follows by induction. $\square$

## RSE-Invertible Variables

Now, we show how to identify mutex groups in $\mathrm{E}_\Pi$ whose counterparts form variables in $\mathcal{P}(\mathrm{E}_\Pi)$ that are RSE-invertible. This condition is required for black variables in the tractable fragment of red-black planning we consider. First, we formulate conditions for RSE-invertibility of facts and mutex groups. Then, we show that, given a fam-group map, these conditions are both sufficient and necessary for RSE-invertibility of the resulting FDR variables.

For the rest of this section, let $E_\Pi = \langle \Pi, \mathcal{G}, \gamma \rangle$ for $\Pi = \langle \mathcal{F}, \mathcal{O}, s_I, s_G \rangle$ denote an MGE-STRIPS planning task, and let $\mathcal{P}(E_\Pi) = \langle \mathcal{V}, \mathcal{O}, \psi_I, \psi_G \rangle$ denote an FDR planning task constructed from $E_\Pi$ according to Definition 3.

**Definition 5.** A fact $f \in \mathcal{F}$ is **RSE-invertible** if

(I1) there exist $o, o' \in \mathcal{O}$ such that $f \in \mathrm{add}(o)$ and $f \in \mathrm{del}(o')$, and

(I2) for every operator $o \in \mathcal{O}$ such that $f \in \mathrm{del}(o)$ it holds that there exists an operator $o' \in \mathcal{O}$ such that $f \in \mathrm{add}(o')$ and $\mathrm{pre}(o') \subseteq \mathrm{pre}(o) \cup \mathrm{add}(o)$, and

(I3) for every operator $o \in \mathcal{O}$ such that $f \in \mathrm{add}(o)$ it holds that there exists an operator $o' \in \mathcal{O}$ such that $f \in \mathrm{del}(o')$ and $\mathrm{pre}(o') \subseteq \mathrm{pre}(o) \cup \mathrm{add}(o)$.

We also call a set of facts (or mutex group) $F \subseteq \mathcal{F}$ RSE-invertible if every $f \in F$ is RSE-invertible.

For the main result of this section (Theorem 9), we need to prove several auxiliary lemmas first. We start by showing that the condition $\mathrm{pre}(o) \subseteq \mathrm{pre}(o') \cup \mathrm{add}(o')$ from (I2-3) corresponds to the RSE-invertibility condition on edges in DTGs as long as we have transitions in both directions.

**Lemma 6.** *Let $G \in \mathcal{G}$ be a mutex group and $o, o' \in \mathcal{O}$ be two operators with edges $(x, q_o, x')$ and $(x', q_{o'}, x)$ in the DTG of $V_G$. Then $\mathrm{pre}(o) \subseteq \mathrm{pre}(o') \cup \mathrm{add}(o')$ iff $\mathrm{pre}(q_o)[\mathcal{V} \setminus V_G] \subseteq \mathrm{pre}(q_{o'})[\mathcal{V} \setminus V_G] \cup \mathrm{eff}(q_{o'})[\mathcal{V} \setminus V_G]$.*

*Proof.* "$\Leftarrow$": It is easy to see that $f \in \mathrm{pre}(o)$ iff $\langle V_{G'}, f \rangle \in \mathrm{pre}(q_o)$, and $f \in \mathrm{pre}(o')$ iff $\langle V_{G'}, f \rangle \in \mathrm{pre}(q_{o'})$, and $f \in \mathrm{add}(o')$ iff $\langle V_{G'}, f \rangle \in \mathrm{eff}(q_{o'})$ for every $G' \in \mathcal{G}$, therefore it holds that $\mathrm{pre}(o) \setminus G \subseteq (\mathrm{pre}(o') \cup \mathrm{add}(o')) \setminus G$. It remains to show that also $\mathrm{pre}(o) \cap G \subseteq (\mathrm{pre}(o') \cup \mathrm{add}(o')) \cap G$ (because, for all $G' \in \mathcal{G}$, $\langle V_{G'}, \perp_{G'} \rangle$ is never explicitly represented as a fact in $\mathrm{pre}(o)$). There are two cases:

(1) If $V_G \in \mathrm{vars}(\mathrm{pre}(q_o))$, then $\mathrm{pre}(o)[V_G] = f = x \in G$, because $(x, q_o, x')$ is an edge in the DTG of $V_G$, therefore $\mathrm{pre}(o) = f$. Moreover, since there is also an edge $(x', q_{o'}, f)$ in the DTG of $V_G$, it follows that $\mathrm{eff}(q_{o'})[V_G] = f$ and therefore $f \in \mathrm{add}(o')$. (2) If $V_G \notin \mathrm{vars}(\mathrm{pre}(q_o))$ or $\mathrm{pre}(q_o)[V_G] = \perp_G$, then $G \cap \mathrm{pre}(o) = \emptyset$.

"$\Rightarrow$": For every $G \in \mathcal{G}$ s.t. $G \cap \mathrm{pre}(o) \neq \emptyset$ there exists $f \in G$ s.t. $f \in \mathrm{pre}(o)$, and $\mathrm{pre}(q_o)[V_G] = f$, and $\mathrm{pre}(q_{o'})[V_G] = f$ or $\mathrm{eff}(q_{o'})[V_G] = f$. So, it remains to show that the claim holds also for every $H \in \mathcal{G}$, $H \neq G$, such that $H \cap \mathrm{pre}(o) = \emptyset$ and $V_H \in \mathrm{vars}(\mathrm{pre}(q_o))$ and therefore $\mathrm{pre}(q_o)[V_H] = \perp_H$. If $\mathrm{pre}(q_o)[V_H] = \perp_H$, then $\gamma(H) \cap \mathrm{pre}(o) \neq \emptyset$. Let $y \in \gamma(H) \cap \mathrm{pre}(o)$. Therefore, $y \in \mathrm{pre}(o')$ or $y \in \mathrm{add}(o')$.

(1) If $y \in \mathrm{pre}(o')$ then $H \cap \mathrm{pre}(o') = \emptyset$, because $\mathrm{pre}(o')$ is not a mutex (S2). Therefore, $\mathrm{pre}(q_{o'})[V_H] = \perp_H$.

(2) If $y \in \mathrm{add}(o')$ then $H \cap \mathrm{add}(o') = \emptyset$ (because $o'[\![\mathrm{pre}(o')]\!]$ is not mutex (S2)) and there exists $y' \in \mathrm{pre}(o') \cap \mathrm{del}(o')$ s.t. $y' \in \gamma(H)$ (because $\gamma(H)$ is a fam-group). (2a) If $y' \in H$, then $\mathrm{eff}(q_{o'})[V_H] = \perp_H$, because $H \cap \mathrm{add}(o') = \emptyset$. (2b) If $y' \in \gamma(H) \setminus H$, then $H \cap \mathrm{pre}(o') = \emptyset$ (because $y' \in \mathrm{pre}(o')$ and $\mathrm{pre}(o') \notin \mathcal{M}_\mathcal{G}$), therefore $\mathrm{pre}(q_{o'})[V_H] = \perp_H$. $\square$

Now, we are ready to prove that RSE-invertibility of mutex groups implies RSE-invertibility of the corresponding

FDR variables, and vice versa. We separate the case of singleton mutex groups, and larger mutex groups, which require the additional condition (S7).

**Lemma 7.** *Let $G = \{f\}$ be a mutex group. Then $f$ is RSE-invertible iff $V_G$ is RSE-invertible.*

*Proof.* "$\Rightarrow$": Since $\mathrm{dom}(V_G) = \{f, \perp_G\}$ and we need to prove that every edge in the DTG of $V_G$ is RSE-invertible, we need to investigate two cases:

(1) Let $(f, q_o, \perp_G)$ be an edge in the DTG of $V_G$. Then $\mathrm{eff}(q_o)[V] = \perp_G$ and either $\mathrm{pre}(q_o)[V] = f$ or $V \notin \mathrm{vars}(\mathrm{pre}(q_o))$. So we have $f \notin \mathrm{add}(o)$ and $f \in \mathrm{del}(o)$, therefore there exists $o'$ s.t. $f \in \mathrm{add}(o')$ and $\mathrm{pre}(o') \subseteq \mathrm{pre}(o) \cup \mathrm{add}(o)$. So, $f \notin \mathrm{pre}(o')$ because $f \in \mathrm{add}(o')$, therefore $\mathrm{pre}(q_o)[V_G] = \perp_G$ or $V_G \notin \mathrm{vars}(\mathrm{pre}(q_{o'}))$, therefore there is an edge $(\perp_G, q_{o'}, f)$. The rest follows from Lemma 6.

(2) Let $(\perp_G, q_o, f)$ be an edge in the DTG of $V_G$. Then $\mathrm{eff}(q_o)[V_G] = f$ and either $\mathrm{pre}(q_o)[V_G] = \perp_G$ or $V \notin \mathrm{vars}(\mathrm{pre}(q_o))$. So, we have $f \in \mathrm{add}(o)$ and therefore $f \notin \mathrm{pre}(o)$ and $f \notin \mathrm{del}(o)$. Therefore, there exists $o' \in \mathcal{O}$ s.t. $f \in \mathrm{del}(o')$ and $\mathrm{pre}(o') \subseteq \mathrm{pre}(o) \cup \mathrm{add}(o)$. Now, to prove that there is an edge $(f, q_{o'}, \perp_G)$ (and therefore this lemma follows from Lemma 6), we need to show that $\mathrm{pre}(q_{o'})[V_G] \neq \perp_G$, i.e., $\mathrm{pre}(q_{o'})[V_G] = f$ or $V \notin \mathrm{vars}(\mathrm{pre}(q_{o'}))$. If $\mathrm{pre}(q_{o'})[V_G] = \perp_G$, then there is $f' \in \gamma(G)$ s.t. $f' \in \mathrm{pre}(o') \cap \mathrm{del}(o')$, because $\gamma(G)$ is a fam-group. Therefore, $\{f, f'\} \subseteq \mathrm{del}(o')$, which is a contradiction because $\{f\} \cup \mathrm{pre}(o')$ is a mutex (S2).

"$\Leftarrow$": If $V_G$ is RSE-invertible, then for every edge $(x, q_o, x')$ in $\mathcal{D}_{V_G}$ there exists $(x', q_{o'}, x)$ such that $\mathrm{pre}(q_{o'})[\mathcal{V} \setminus V_G] \subseteq \mathrm{pre}(q_o)[\mathcal{V} \setminus V_G] \cup \mathrm{eff}(q_o)[\mathcal{V} \setminus V_G]$. Therefore, it follows from Lemma 6 that also $\mathrm{pre}(o') \subseteq \mathrm{pre}(o) \cup \mathrm{add}(o)$. Moreover, from the construction (T2b) it follows that if $x' = f$, then $f \in \mathrm{add}(o)$ and $f \in \mathrm{del}(o')$, and vice versa, if $x = f$, then $f \in \mathrm{del}(o)$ and $f \in \mathrm{add}(o')$. Therefore, the conditions (I1-3) are satisfied. $\square$

A mutex group $M \in \mathcal{G}$ such that $|M| \geq 2$ requires the fam-group map $\gamma$ to map $M$ to its superset fam-group (S7). Recall the example in Figure 1. The facts $f_1$ and $f_2$ are RSE-invertible, but $f_3$ and $f_4$ are not, because of the operator $o_{34}$. So, we would like to construct an RSE-invertible variable from $G = \{f_1, f_2\}$. If we set $\gamma(G) = \emptyset$, then the resulting DTG (Figure 1c) will have edges between $f_1$ and $f_2$ that are not RSE-invertible, because the operator $o_{31}$ has $f_3$ as its precondition, which is not part of $\mathrm{pre}(o_{42}) \cup \mathrm{add}(o_{42})$. However, $\gamma(G) = F$ results in the RSE-invertible DTG (Figure 1b).

**Lemma 8.** *Let $G \in \mathcal{G}$ be a mutex group with $|G| \geq 2$ and $\gamma(G) \neq \emptyset$. Then $G$ is RSE-invertible iff $V_G$ is RSE-invertible.*

*Proof.* "$\Rightarrow$": Since $\gamma(G)$ is a fam-group s.t. $G \subseteq \gamma(G)$ and $|G| \geq 2$ and (S6), for every operator $o \in \mathcal{O}$ and every $f \in G$ it holds that $f \in \mathrm{del}(o)$ implies $f \in \mathrm{pre}(o)$. Let $(d, q_o, d')$ denote an edge in the DTG of $V_G$. Then $\mathrm{eff}(q_o)[V_G] = d'$ and either $\mathrm{pre}(q_o)[V_G] = d$ or $V_G \notin \mathrm{vars}(\mathrm{pre}(q_o))$. Now, if we show that there also exist an edge $(d', q_{o'} d)$ for some

$o' \in \mathcal{O}$ s.t. $\mathrm{pre}(o') \subseteq \mathrm{pre}(o) \cup \mathrm{add}(o)$, then the rest follows from Lemma 6. So three cases need to be investigated.

(1) If $d, d' \in G$, then $d' \in \mathrm{add}(o)$. And since $\gamma(G)$ is a fam-group, we have $V_G \in \mathrm{vars}(\mathrm{pre}(q_o))$ and $d \in \mathrm{pre}(o) \cap \mathrm{del}(o)$. Therefore, there exists $o' \in \mathcal{O}$ s.t. $d \in \mathrm{add}(o')$ and $\mathrm{pre}(o') \subseteq \mathrm{pre}(o) \cup \mathrm{add}(o)$. Thus, $d' \in \mathrm{pre}(o')$, because $d \notin \mathrm{pre}(o')$ and $\gamma(G) \cap (\mathrm{pre}(o) \cup \mathrm{add}(o)) = \{d, d'\}$. Therefore, there exists an edge $(d', q_{o'}, d)$.

(2) If $d \in G$ and $d' = \perp_G$, then $\mathrm{add}(o) \cap G = \emptyset$ and $\mathrm{del}(o) \cap G \neq \emptyset$, therefore $d \in \mathrm{del}(o)$, because otherwise there would exist $x \in G$, $x \neq d$, s.t. $x \in \mathrm{del}(o)$, therefore $x \in \mathrm{pre}(o)$, therefore $\mathrm{pre}(q_o)[V_G] = x$ which is in contradiction. So we have $d \in \mathrm{del}(o)$ and also $d \in \mathrm{pre}(o)$. Therefore, there exists $o'$ s.t. $d \in \mathrm{add}(o')$ and $\mathrm{pre}(o') \subseteq \mathrm{pre}(o) \cup \mathrm{add}(o)$. So since $d \in \mathrm{pre}(o)$ and $\mathrm{add}(o) \cap G = \emptyset$, then $\mathrm{pre}(o') \cap G = \emptyset$, therefore we have $\mathrm{eff}(q_{o'})[V_G] = d$ and either $\mathrm{pre}(q_{o'})[V_G] = \perp_G$ or $V_G \notin \mathrm{vars}(\mathrm{pre}(q_{o'}))$. Therefore, we have an edge $(\perp_G, q_{o'}, d)$.

(3) If $d = \perp_G$ and $d' \in G$, then $d' \in \mathrm{add}(o)$, therefore there exists $f \in \gamma(G) \setminus G$ s.t. $f \in \mathrm{pre}(o) \cap \mathrm{del}(o)$. Thus, there exists $o' \in \mathcal{O}$ s.t. $f \in \mathrm{add}(o')$ and $\mathrm{pre}(o') \subseteq \mathrm{pre}(o) \cup \mathrm{add}(o)$. Hence, $d' \in \mathrm{pre}(o') \cap \mathrm{del}(o')$ because $\gamma(G) \cap \mathrm{pre}(o') \cap \mathrm{del}(o') \neq \emptyset$ ($\gamma(G)$ is a fam-group) and $(\mathrm{pre}(o) \cup \mathrm{add}(o)) \cap \gamma(G) = \{f, d'\}$ and $f \notin \mathrm{pre}(o')$. Therefore, $\mathrm{pre}(q_{o'})[V_G] = d'$ and $\mathrm{eff}(q_{o'})[V_G] = \perp_G$, and thus we have $(d', q_{o'}, \perp_G)$.

"$\Leftarrow$": Let $(x, q_o, x')$ denote an edge in $\mathcal{D}_{V_G}$. If $x' \in G$, then $x' \in \mathrm{add}(o)$ (T2b). Since $\gamma(G)$ is a fam-group, either $\mathrm{pre}(q_o)[V_G] = \perp_G$, or $\mathrm{pre}(q_o)[V_G] = x$ s.t. $x \in G$. Therefore if $x \in G$, then $x \in \mathrm{pre}(o) \cap \mathrm{del}(o)$. Therefore, for every such $q_o$ it holds that either $G \cap \mathrm{del}(o) \neq \emptyset$ or $G \cap \mathrm{add}(o) \neq \emptyset$. Since $V_G$ is RSE-invertible, for every edge $(x, q_o, x')$ there exist an edge $(x', q_{o'}, x)$ s.t. $\mathrm{pre}(q_{o'})[\mathcal{V} \setminus V_G] \subseteq \mathrm{pre}(q_o)[\mathcal{V} \setminus V_G] \cup \mathrm{eff}(q_o)[\mathcal{V} \setminus V_G]$, therefore it follows from Lemma 6 that $\mathrm{pre}(o') \subseteq \mathrm{pre}(o) \cup \mathrm{add}(o)$. Hence, the conditions (I1-3) are satisfied. $\square$

**Theorem 9.** *Let $G \in \mathcal{G}$ be a mutex group with $|G| = 1$ or $\gamma(G) \neq \emptyset$. Then $G$ is RSE-invertible iff $V_G$ is RSE-invertible.*

*Proof.* It follows directly from Lemma 7 and Lemma 8. $\square$

## Causal Graph

Besides RSE-invertibility of black variables, the tractable fragment of red-black planning requires the black causal graph to be acyclic. To achieve that, we show how to find a causal link between mutex groups that is translated into FDR.

**Definition 10.** Given a mutex group $G \in 2^{\mathcal{F}}$, two distinct facts $f \in G$ and $f' \notin G$, and a fam-group map $\gamma$ on $2^{\mathcal{F}}$, we say that there is a **causal link from** $(G, f)$ **to** $f'$ iff there exists an operator $o \in \mathcal{O}$ such that (i) $f' \in \mathrm{del}(o) \cup \mathrm{add}(o)$, and (ii) $f \in \mathrm{pre}(o) \cup \mathrm{del}(o) \cup \mathrm{add}(o)$, or $\gamma(G) \cap \mathrm{pre}(o) \neq \emptyset$.

**Theorem 11.** *Let $\mathrm{E}_\Pi = \langle \Pi, \mathcal{G}, \gamma \rangle$ be an MGE-STRIPS task with operators $\mathcal{O}$, $A, B \in \mathcal{G}$ be two distinct mutex groups, and $\mathrm{CG}_\mathcal{V}$ be the causal graph of $\mathcal{P}(\mathrm{E}_\Pi)$. There is an edge $(V_A, V_B)$ in $\mathrm{CG}_\mathcal{V}$ iff there exist $f \in A$ and $f' \in B$ such that there is a causal link from $(A, f)$ to $f'$.*

*Proof.* "$\Leftarrow$": Let $o \in \mathcal{O}$ be the operator inducing the causal link from $(A, f)$ to $f'$. If $f \in \mathrm{pre}(o)$, then $\mathrm{pre}(q_o)[V_A] = f$. If $f \in \mathrm{add}(o)$, then $\mathrm{eff}(q_o)[V_A] = f$. If $f \in \mathrm{del}(o)$, then either $\mathrm{eff}(q_o)[V_A] = \perp_A$ or there exist $x \in A$ s.t. $\mathrm{eff}(q_o)[V_A] = x$. If $\gamma(A) \neq \emptyset$ and $\gamma(A) \cap \mathrm{pre}(o) \neq \emptyset$, then either $\mathrm{pre}(q_o)[V_A] = x$ for some $x \in A$, or $\mathrm{pre}(q_o)[V_A] = \perp_A$. Therefore, $V_A \in (\mathrm{vars}(\mathrm{pre}(q_o)) \cup \mathrm{vars}(\mathrm{eff}(q_o))$, and for similar reasons $V_B \in \mathrm{vars}(\mathrm{eff}(q_o))$.

"$\Rightarrow$": If $(V_A, V_B)$ in $\mathrm{CG}_{\{V_A, V_B\}}$, then there exists $q_o \in \mathcal{O}$ s.t. $V_A \in (\mathrm{vars}(\mathrm{pre}(q_o)) \cup \mathrm{vars}(\mathrm{eff}(q_o)))$ and $V_B \in \mathrm{vars}(\mathrm{eff}(q_o))$. If $\mathrm{pre}(q_o)[V_A] \in A$ or $\mathrm{eff}(q_o)[V_A] \in A$, and $\mathrm{eff}(q_o)[V_B] \in B$, then it follows trivially that $A \cap (\mathrm{pre}(o) \cup \mathrm{add}(o)) \neq \emptyset$ and $B \cap \mathrm{add}(o) \neq \emptyset$. So what remains is to show that (i) if $\mathrm{pre}(q_o)[V_A] = \perp_A$, then $\gamma(A) \neq \emptyset$ and $\gamma(A) \cap \mathrm{pre}(o) \neq \emptyset$ (T2a); (ii) if $\mathrm{eff}(q_o)[V_A] = \perp_A$, then $A \cap \mathrm{del}(o) \neq \emptyset$ (T2b); (iii) if $\mathrm{eff}(q_o)[V_B] = \perp_B$, then $B \cap \mathrm{del}(o) \neq \emptyset$ (T2b). $\square$

Theorem 11 shows that causal graphs in the FDR planning task $\mathcal{P}(\mathrm{E}_\Pi)$ correspond exactly to the causal links in $\mathrm{E}_\Pi$ as per Definition 10. So, extracting *acyclic* causal graphs needed for the tractable fragment of red-black planning require to find subsets of mutex groups consisting of facts whose causal links form an acyclic graph. In the next section, we show how to find such subsets of mutex groups.

## Inference of Black Variables

We have shown how to find RSE-invertible mutex groups and what conditions must be met in order to form a causal link between FDR variables created from mutex groups. Now, we put everything together and show how to select mutex groups that can be translated into RSE-invertible FDR variables that form an acyclic causal graph, and thus can be painted black.

**Definition 12.** Given a STRIPS planning task $\Pi$ with facts $\mathcal{F}$, a fam-group map $\gamma$ over $2^{\mathcal{F}}$, and a set of mutex groups $\mathcal{K} \subseteq 2^{\mathcal{F}}$ such that for every $G \in \mathcal{K}$ it holds that either $|G| = 1$, or $\gamma(G) \neq \emptyset$, $\mathcal{X}(\Pi, \mathcal{K}, \gamma) = (V, E)$ denotes a digraph with vertices $V \subseteq \mathcal{K} \times \mathcal{F}$ and edges $E \subseteq V \times V$, where $(G, f) \in V$ iff $G \in \mathcal{K}$ and $f \in G$, and $((G, f), (G', f')) \in E$ iff $f = f'$ or there is a causal link from $(G, f)$ to $f'$. Given a cycle $c$ in $\mathcal{X}(\Pi, \mathcal{K})$, let $\mathcal{C}(c) = \{G \mid (G, f) \in c\}$.

**Theorem 13.** *Let $\mathrm{E}_\Pi = \langle \Pi, \mathcal{G}, \mathcal{M}_\mathcal{G} \rangle$ be an MGE-STRIPS planning task, $\mathcal{K} \subseteq \mathcal{G}$ be a set of RSE-invertible mutex groups such that $|G| = 1$ or $\gamma(G) \neq \emptyset$ for every $G \in \mathcal{K}$, and $\mathcal{V}_\mathcal{K}$ be the set of corresponding variables from $\mathcal{P}(\mathrm{E}_\Pi)$. Then $\mathrm{CG}_{\mathcal{V}_\mathcal{K}}$ is acyclic iff for every cycle $c$ in $\mathcal{X}(\Pi, \mathcal{K}, \gamma)$ it holds that $|\mathcal{C}(c)| = 1$.*

*Proof Sketch.* It follows directly from Theorem 11, because if $|\mathcal{C}(c)| = 1$, then every cycle of causal links is contained within one mutex group. $\square$

Algorithm 1 encapsulates the algorithm for inference of a set of RSE-invertible mutex groups $\mathcal{K}$, and a fam-group map $\gamma$, that can be translated into red-black planning task as black variables. We assume we are given a STRIPS planning task $\Pi$, and a set of fam-groups $\mathcal{H} \subseteq 2^{\mathcal{F}}$ that can be inferred by one of the algorithm proposed by Helmert (2009), Fišer and

**Algorithm 1:** Inference of RSE-invertible mutex groups forming acyclic causal graph.

---

**Input:** A STRIPS planning task $\Pi = \langle \mathcal{F}, \mathcal{O}, s_I, s_G \rangle$, a set of fam-groups $\mathcal{H} \subseteq 2^{\mathcal{F}}$, a set of RSE-invertible facts $\mathcal{N} \subseteq \mathcal{F}$.

**Output:** A set of RSE-invertible mutex groups $\mathcal{K}$, a fam-group map $\gamma$ over $\mathcal{K}$

1 $\mathcal{H}' \leftarrow \{\{f\} \mid f \in \mathcal{N} \setminus \bigcup_{G \in \mathcal{H}} G\}$;
2 Construct a fam-group map $\gamma'$: $\gamma'(G) = \emptyset$ for every $G \in \mathcal{H}'$, and $\gamma'(G \cap \mathcal{N}) = G$ for every $G \in \mathcal{H}$ s.t. $G \cap \mathcal{N} \neq \emptyset$;
3 $\mathcal{H}^{\star} \leftarrow \mathcal{H}' \cup \{G \cap \mathcal{N} \mid G \in \mathcal{H}, G \cap \mathcal{N} \neq \emptyset\}$;
4 Construct the digraph $\mathfrak{X}(\Pi, \mathcal{H}^{\star}, \gamma') = (V, E)$.;
5 Find vertices $V' \subseteq V$ such that for every cycle $c$ in the subgraph of $\mathfrak{X}(\Pi, \mathcal{H}^{\star}, \gamma)$ induced by $V'$ it holds that $|\mathcal{C}(c)| = 1$;
6 $Z \leftarrow \{(G, \{f \mid (G, f) \in V'\}) \mid (G, f') \in V'\}$.;
7 Construct a fam-group map $\gamma$: For every $(G, X) \in Z$, set $\gamma(X) = \emptyset$ if $|X| = 1$ and $\gamma(X) = G$ otherwise;
8 $\mathcal{K} \leftarrow \{X \mid (G, X) \in Z\}$;

---

Komenda (2018), or Fišer (2020). In steps 1-3, we prepare RSE-invertible mutex groups and a mapping to the corresponding fam-groups. In step 1, $\mathcal{H}'$ is constructed as a set of singletons, each corresponding to the RSE-invertible fact that is not covered by any input fam-group, i.e., these facts can be translated only to binary FDR variables. In step 2, the fam-group map is constructed in the following way: (i) each singleton from $\mathcal{H}'$ is mapped to an empty set, and (ii) for each fam-group $G$ containing at least one RSE-invertible fact, we map its RSE-invertible subset $G \cap \mathcal{N}$ to $G$. In step 3, $\mathcal{H}^{\star}$ is constructed as a set of RSE-invertible mutex groups covering all RSE-invertible facts. In step 4, the graph defined in Definition 12 is constructed, and in step 5, we find a subset of vertices having cycles only within each mutex group. Finally, in the last three steps we extract the solution.

Note that the step 5 is a variant of an NP-complete feedback vertex set problem (Karp 1972). We decided to use a naive approach and solve this problem by an integer linear program (ILP), where each vertex from $\mathfrak{X}(\Pi, \mathcal{H}^{\star}, \gamma')$ corresponds to a binary variable, and each constraint corresponds to a cycle that has to be avoided. Instead of listing all cycles, we first add all cycles between pairs of vertices from different mutex groups, then we solve the problem, and check whether the resulting solution has a cycle. If we find a cycle, we add it as another constraint, solve the problem, and continue in this manner until we find a feasible solution.

As objective function of the ILP, we tried two variants. The first variant is simply a maximization of the number of facts in the resulting mutex groups. The second variant weights facts using the number of *conflicts* in a relaxed plan $\pi^{+}$ obtained from the FF heuristic (Hoffmann and Nebel 2001) computed on the STRIPS representation. Following prior work (Domshlak, Hoffmann, and Katz 2015), we say a fact $f$ has a conflict if there is an operator $o$ in $\pi^{+}$ such that $f \in \text{pre}(o)$ and $f$ is not satisfied when executing $\pi^{+}$ with the non-relaxed semantics. For each mutex group $M$, we sum the number of conflicts from all $f \in M$, and we set

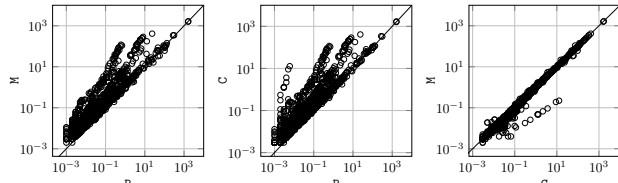

Figure 2: Per-task comparison of the translation time (in seconds).

the weight of each fact $f \in M$ as the sum over $|M|$. This way we prioritize mutex groups containing facts with most conflicts in the relaxed plan.

Note that Algorithm 1 describes only the inference of RSE-invertible variables. The rest of the FDR variables, i.e., the red variables, are constructed from the remaining facts not covered by the resulting black variables by greedy approach described by Helmert (2009), i.e., we greedily maximize the size of red variables.

## Experimental Evaluation

We implemented Algorithm 1 in C, used fam-groups inferred from PDDL (Fišer 2020)[2], and used the red-black heuristic $h^{\text{RB}}$ (Domshlak, Hoffmann, and Katz 2015) implemented in FD (Helmert 2006). The experiments were performed on Intel(R) Xeon(R) Scalable Gold 6146 machines with CPLEX solver v12.6, and time and memory limits of 30min and 8GiB. Out of all benchmarks from International Planning Competitions of 1998 to 2018 satisficing planning tracks, we selected all tasks where no conditional effects were created and at least one variable could be painted black.

We compare the following configurations: the baseline configuration B runs the painting strategy denoted as $\mathbf{A}$[3] by Domshlak, Hoffmann, and Katz (2015) on the default FDR encoding (Helmert 2009); M maximizes the number of black facts using Algorithm 1; C is the variant of Algorithm 1 using conflicts in a relaxed plan; and the "oracle" O picks the best FDR encoding and painting for each tested task. It considers B, and five best encodings (by objective value) from each of M and C, which we obtain by repeatedly solving the corresponding ILP, each time disallowing previous solutions by additional constraints. The encodings are chosen in terms of coverage, preferring tasks solved in the initial state by $h^{\text{RB}}$. O serves to illustrate the potential of our customized encodings, assuming perfect knowledge about which encoding works best for a given task.

The most time-demanding part of Algorithm 1 is solving the ILP (step 5). The average and median time spent in this step was 3.3 seconds and 12 milliseconds, respectively, for both C and M. The maximum was more than six minutes, but it took more than a minute only for 15 tasks in transport, 7 tasks in visitall, and 1 task in the satellite domain. Figure 2 shows a comparison of running times (in seconds) of

---

[2]https://gitlab.com/danfis/cpddl, branch ijcai21-fdr-red-black
[3]The painting strategy greedily paints variables red until the black causal graph is acyclic, preferring keeping variables with larger number of incident edges to black variables.

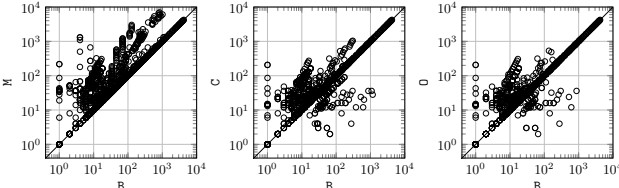

Figure 3: Per-task comparison of the number of black FDR facts.

| domain | coverage | | | | solved initial state | | | | sgl |
|---|---|---|---|---|---|---|---|---|---|
| | B | M | C | O | B | M | C | O | |
| barman11 (20) | **9** | 2 | 3 | **9** | 0 | 0 | 0 | 0 | 0 |
| blocks00 (35) | **35** | 33 | 32 | **35** | 0 | 1 | **2** | **2** | 0 |
| depot02 (22) | **17** | 14 | 14 | **17** | 1 | 1 | 1 | 1 | 22 |
| driverlog02 (20) | 19 | 19 | 19 | **20** | 1 | 1 | 1 | **2** | 0 |
| elevators08/11 (50) | **50** | 41 | **50** | **50** | 50 | 2 | 50 | 50 | 0 |
| floortile11 (20) | **8** | **8** | 7 | **8** | 0 | 0 | 0 | 0 | 20 |
| hiking14 (20) | 13 | 16 | 16 | **20** | 0 | 0 | 0 | 0 | 0 |
| mystery98 (28) | 18 | 18 | **19** | **19** | 0 | 0 | 0 | 0 | 0 |
| pipesw-notank04 (40) | **24** | 23 | 23 | **24** | 0 | 0 | 0 | 0 | 30 |
| pipesw-tank04 (40) | 16 | 16 | 16 | **18** | 0 | 0 | 0 | 0 | 0 |
| satellite02 (36) | 36 | 36 | 36 | 36 | 10 | 10 | 14 | **21** | 2 |
| scanalyzer08/11 (35) | 31 | **35** | **35** | **35** | 0 | 0 | 0 | 0 | 0 |
| sokoban08/11 (50) | 46 | 46 | **48** | **48** | 0 | 0 | 0 | 0 | 23 |
| storage06 (30) | 19 | 19 | 19 | **21** | 3 | 3 | 3 | 3 | 0 |
| termes18 (20) | 14 | 14 | 14 | **16** | 0 | 0 | 0 | 0 | 0 |
| tidybot11 (20) | 16 | 15 | 16 | **18** | 0 | 0 | 0 | 0 | 0 |
| transport08/11/14 (70) | **70** | 65 | **70** | **70** | 70 | 0 | 70 | 70 | 0 |
| trucks06 (30) | 15 | 16 | 16 | **17** | 0 | 0 | 0 | 0 | 0 |
| others (647) | 497 | 497 | 497 | 497 | 172 | 172 | 172 | 172 | 428 |
| $\Sigma$ (1 233) | 953 | 933 | 950 | **978** | 307 | 190 | 313 | **321** | 525 |

Table 1: Number of tasks solved (left), number of tasks solved in the initial state (center), and number of tasks with a single maximal set of black variables (right). B, M, C, and O are described in the text. Domains with identical results are summarized in "others".

the whole translation process. Although Algorithm 1 often requires more time in comparison to the greedy approach of the baseline, it, for most cases, leaves enough time for the planner, as we describe below.

The scatter plots in Figure 3 show that our methods are able to increase the number of black facts by several orders of magnitude. The oracle often uses significantly less black facts than B, indicating that it is not necessarily beneficial to have more black facts. In fact, having more black variables (facts) can negatively impact heuristic computation time, without improving informativeness.

Table 1 shows the number of tasks solved (coverage) in the left, and the number of tasks solved by $h^{\text{RB}}$ in the initial state in the center. B clearly outperforms M which, again, shows that maximizing the number of black facts is not necessarily the best painting strategy. C and B are very similar in coverage, with strengths in different domains. C can solve six more tasks in the initial state resulting from only two domains, blocks and satellite.

The oracle variant O demonstrates the potential of our approach if we better understand how to select the best encoding for a given task. B already solves a large portion of tasks, but here we show that it is still possible to create an encoding and painting that could further increase this number by at least 25 tasks, and 14 more can be solved in the initial state. Unfortunately, it is unclear how to directly compare the quality of individual encodings for a given task without running the search. Maximizing the number of black facts turned out to be too simplistic as an optimization criterion. Considering conflicts in a relaxed plan improves the results in some domains, but it is still unclear what properties identify a good FDR encoding and painting for red-black planning.

The rightmost column in Table 1 shows the number of tasks with a single maximal set of black variables. That is, it shows the number of tasks where different paintings can only be achieved by painting black a subset of the facts that were already painted black by C and M. In other words, both C and M created exactly the same set of black variables, but B could paint only less variables black. This is exactly what happened in the depot domain where painting less variables black proved to generate a more suitable FDR for the red-black heuristic.

## Conclusion

We devised a method to custom-design FDR encodings of classical planning tasks, showcasing how a tractable fragment of red-black planning can be formulated into the translation from PDDL to FDR. By describing this translation as an integer linear program, we can explore the space of possible tractable red-black FDR encodings. Our evaluation shows an increase in the number of black facts by several orders of magnitude. Our approach is flexible, allowing to express complex properties based on, e.g., DTGs or causal graphs. Thus, custom-designed FDR encodings have the potential of benefiting planning techniques beyond red-black planning, opening up a wide range of possibilities for future research.

## Acknowledgements

Daniel Gnad was supported by the German Research Foundation (DFG), under grant HO 2169/6-2, "Star-Topology Decoupled State Space Search". Jörg Hoffmann's research group has received support by DFG grant 389792660 as part of TRR 248 (see perspicuous-computing.science).

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
