# OpenReview forum: "Custom-Design of FDR Encodings: The Case of Red-Black Planning"
_icaps-conference.org/ICAPS/2021/Workshop/HSDIP — HSDIP 2021_

### Official Review · AnonReviewer2 · 2021-05-26

**Confidence:** 3
**Overall Score:** Accept

**Review:**

This paper presents an approach to computing finite domain representations (FDR) that are well suited for red-black planning. The underlying idea is not to blindly translate a pddl description into FDR, but to choose an encoding that ensures an tractable red-black FDR encoding.The paper describes how such an encoding can be found and proves that this method is correct.  In addition, an empirical evaluation is conducted comparing an approach that attempts to generate as many black variables as possible with a red-black configuration without this particular design of FDR. While the results of this comparison are mixed, an oracle selector over multiple FDR encodings shows the potential of the underlying idea.

The newly presented approach is well motivated and breaks out of the classical pattern of taking an FDR encoding as given without optimising it for the subsequent search performed. Overall, the topic of the paper fits the workshop and is well written. However, the theory section introduces a lot of definitions and theoretical results that are not so easy to follow if you are not familiar with all kinds of concepts of classical planning. Perhaps a little more explanation or intuition would be appropriate here. In my opinion, especially a small example on RSE-invertibility in the background section could help a reader who is not quite familiar with this concept, as it plays a central role throughout the paper.

Otherwise, I have only some minor comments and questions concerning the empirical evaluation, which I list below.

Regarding the computation of the FDR encodings, it would be interesting to know how long it took to compute them with the ILP solver. Are these times included in the 30 min? I assume that these were precomputed for the oracle?

What kind of search was carried out, greedy best-first search with lazy evaluation (maybe I missed it)?

As far as I understand, the oracle configuration chooses the best FDR encoding by selecting one of up to 11 encodings. I wonder if it could be beneficial to use different FDR encodings for heuristic computation in a single search, e.g. by using greedy search with multiple open lists. At least it could be interesting to use all 11 encodings for solving the initial state depending on the time consumption. Do you have an intuition for this?

Overall, the paper presents an interesting approach to classical planning that interleaves the translation and search phases to find a good FDR encoding for red-black planning. In my opinion, this is a valuable contribution, so I recommend accepting the paper.

---

> ### Author Response · Authors · 2021-06-04
> **Response**
>
> Thanks for your review.
>
> As mentioned above, we attached a camera-ready version of the paper from the conference submission as a supplementary material.
>
> To address the raised questions:
>
> 1. The computation times for solving ILP is included in the 30 minutes time limit -- you can see a more detailed analysis of the computation time in the supplementary material. For oracle, we run multiple configurations (each with the same time and memory limits) and then select the best variant.
>
> 2. We ran greedy best-first search with lazy evaluation on the unit-cost transformed task.
>
> 3. Regarding the suggestion of using multiple open lists, each corresponding to a different FDR encoding: We think it might be too computationally demanding to compute red-black plans for all encodings in all states. However, we think it might be feasible to run the red-black planner for all encodings only for the initial state, and, based on the results, choose only one of the encodings. This might get us closer to the oracle variant.

---

### Official Review · AnonReviewer1 · 2021-05-26

**Confidence:** 4
**Overall Score:** Accept

**Review:**

I already reviewed this paper for a conference and since it did not change much
(as far as I can tell) I will for the most part repeat my previous review:

The paper presents an alternative translation of a STRIPS task into an FDR task
which aims to maximize the number of so-called RSE-invertible variables. This
is advantageous for red-black planning since only such variables can be painted
black.

The proposed encoding is theoretically shown to be correct and experimentally
shown to be promising if the best possible painting strategy is used. However,
it is still unclear how to actually choose which variables should be painted
black: the best possible strategy was determined with an oracle, and simply
maximizing the number of black variables performs poorer than the baseline.
Nevertheless, the authors present a different feasible strategy which
comparable or even a bit stronger than the baseline, making the approach
already practically useful as is.

While the experimental results might not seem very significant yet since the
best feasible configuration is comparable to the baseline, I believe that it
offers a strong foundation, and investigating better painting strategies could
yield a significant performance increase. To the best of my knowledge this is
also the first paper to consider alternative FDR encodings specifically
tailored for red-black planning and I think exploring the space of FDR
encodings is an interesting and under-researched idea.

The paper is technically sound as far as I can tell and is very thorough in
its theoretical analysis. It does however suffer from clarity issues: The
theoretical part is very proof-heavy and the proofs are mechanical and hard to
follow. I particularly want to point out the proof to Lemma 6, since it has many
nested case distinctions and technical details. I don't have an concrete
suggestion on how to improve clarity, but given that you have space to spare I
think it could be helpful to add more "filler text" explaining what we are
currently doing and why.

Overall I clearly recommend to accept the paper: the theoretical results are
very convincing and offer a strong foundation for investigating different
painting strategies, and the experimental evaluation shows that the approach
has potential for significantly improving the current state-of-the-art in
red-black planning.

Questions to the authors:
1) How is the final FDR task constructed? Algorithm 1 provides a set of FDR
variables that can be painted black, but how are the rest of the facts split up
into FDR variables, in particular those that are part of the fam-groups
important for the RSE-invertible variables?
2) I can't follow the proof sketch for proposition 4. I do think the encoding
is correct, but the proof sketch states $(V_G, \bot_G) \in s_I$ iff $s_i \in
\mathcal M_{G}$. Doesn't the second part mean that $s_I$ contains a mutex, i.e.
$s_I$ is unreachable (from $s_I$)? Shouldn't it rather be $G \cap s_I = \emptyset$?
The same applies to $pre(o) \in \mathcal M_{G}$.

Minor comments:
- background: "[see Domshlak, Hoffmann, and Katz, 2015]" -> (see ...)
- background: It would be helpful to add a sentence about the intuition of
RSE-invertible variables (why is this property useful/interesting?)
- translation from STRIPS to FDR, first paragraph: It is unclear to me how you
can preserve a DTG structure of black variables when you don't yet have
variables or a painting. I guess you mean the DTG structure of all inferred
mutex groups?
- it is confusing that Definition 2 includes the fam-group map but has no
constraints on it or even any explanation on why it is there.
- proof to Lemma 6: In the first paragraph rather use $G'$ (for example "$f \in
\text{pre}(o)$ iff $(V_G',f) \in \text{pre}(q_o)$") to make clear that this is not the $V_G$
mentioned in the lemma. You could also elaborate on why facts $(V_G',\bot_G')$
don't pose any problems, since they don't appear in the STRIPS representation
and its unclear if they can break something.
- end of proof to Lemma 6: It was at first not clear to me why $H \cap \text{pre}(o') =
\emptyset$ is true here. The argument is that $y' \in \text{pre}(o')$and $y' \notin H$,
i.e. we have a precondition on the fam group thats not in H, thus we can't have
an additional precondition from H (since only one element from the fam group
can be true). Correct?
- Paragraph before Lemma 8: Does every mutex group always have a a superset
fam-group? I assume it must (since otherwise you can't satisfy the condition on
the fam-group map), but why is this the case? As far as I understood fam-groups
are a special case of mutex groups, so not every mutex group is a fam-group
correct?
- Proof to Lemma 8: Does $f \in \text{del}(o)$ always imply $f \in \text{pre}(o)$? The definition
of fam-group says that we need to explicitly delete (i.e. $f \in \text{pre}(o) \cap
\text{del}(o)$) at least as many facts as we add. But lets say we add $f' \in G$ and
explicitly delete $f'' \in G$. Then we could have $f \in \text{del}(o) \setminus \text{pre}(o)$
without violating the fam-group definition.
- Dvorak et al: page numbers missing and the venue seems wrong? I only found a
paper with this title from the KEPS 2013 workshop.

---

> ### Author Response · Authors · 2021-06-04
> **Response**
>
> Thanks for your review.
>
> We attached the (anonymized) camera-ready from our conference submission (as a supplementary material) which should address all of the issues.
>
> In particular, to address the two raised questions:
> 1. The red variables are constructed from the facts that were not used for black variables in the same way as it is done in the translator of Fast Downward: We use a greedy algorithm that tries to allocate as big variables as possible.
>
> 2. You are correct -- this has been fixed in the camera-ready.

---

> > ### Comment · AnonReviewer1 · 2021-06-08
> > **Thank you for your response**
> >
> > Thank you for your answers and for adding the CRC. I agree that they address the minor issues I raised and I particularly liked the explanation why RSE-invertibility is useful.

---

### Decision · Program_Chairs · 2021-06-10

**Decision:**

Accept

**Comment:**

Congratulations, both reviewers agree that the paper is a clear accept, and the provided CRC already fixes any issues raised by the reviews.